# Spatial-temporal differences and influencing factors of coupling coordination between urban quality and technology innovation in the Guangdong-Hong Kong-Macao Greater Bay Area

**Zhichen Yang[1], Yuxi Wu[2], Fangfang Wang[3]\*, Aichun Chen[4], Yixuan Wang[5]**

**1** School of Economics, Jinan University, Guangzhou, China, **2** School of Economics and Management, Beijing University of Technology, Beijing, China, **3** School of Digital Economics, Guangdong University of Finance and Economics, Foshan, China, **4** School of Cultural Tourism and Geography, Guangdong University of Finance and Economics, Guangzhou, China, **5** School of International Economics and Trade, Guangdong Baiyun University, Guangzhou, China

\* wff@gdufe.edu.cn

**Data Availability Statement:** (1)The data in this study are collected in the National Bureau of

## Abstract

The coordinated development of urban quality and technology innovation is an important element of China's technology innovation development strategy in the new era. Based on entropy TOPSIS, coupling coordination models, the gravity center and standard deviation ellipse method, the geographic probe, the GWR, and other methods, we explore the spatial variation and influencing factors of the coupling coordination relationship between urban quality and technology innovation in the Guangdong-Hong Kong-Macao Greater Bay Area from 2011 to 2020. It is found that: (1) the spatial distribution of the coupling coordination shows a characteristic of "high in the middle and low in the surroundings," and (2) the level of benign interaction in the central region is becoming more prominent. The center of gravity of coupling coordination moves toward the northeast, and the standard deviation ellipse shows a contraction trend away from the southwest. (3) Agglomeration capacity, human capital, cultural development, and infrastructure can significantly drive the improvement of the coupling coordination of urban quality and technology innovation, and the two-factor influence is significantly increased after the interaction. (4) The feedback effects of the coupling and coordination states of different cities on each factor have significant spatial differences and show the characteristics of hierarchical band distribution.

## Introduction

As global competition intensifies and the local regional development characteristics of technology innovation are highlighted, innovation-driven urban quality improvement has become the strategic orientation of China's cities in the knowledge economy framework. In 2021, China's

Statistics http://www.stats.gov.cn/ (accessed on 12 February 2022); (2)Statistics bureau of Guangdong province http://stats.gd.gov.cn/ (accessed on 12 February 2022); (3) Census and Statistics Department of the Government of the Hong Kong Special Administrative Region https://www.censtatd.gov.hk/sc/ (accessed on 12 February 2022); (4) Statistics and Population Census Service of the Macao Special Administrative Region Government https://www.dsec.gov.mo/zh-MO/ (accessed on 12 February 2022); (5) The World Bank https://www.worldbank.org/en/home (accessed on 12 February 2022); (6) Web of Science https://www.webofscience.com (accessed on 12 February 2022); (7) China National Intellectual Property Administration https://www.cnipa.gov.cn/ (accessed on 12 February 2022).

**Funding:** This study was supported by the National Social Science Fund of China in the form of a grant [20BJY063], the Guangdong Philosophy and Social Sciences Innovation Project 2022 Special Commissioned Project in the form of a grant [GD22TWCXGC14], and the Key Laboratory of Philosophy and Social Sciences in Guangdong Province of Guangdong University of Finance and Economics in the form of a grant [123-KYJ02022001] to FW.

**Competing interests:** The authors have declared that no competing interests exist.

urbanization rate had reached 64.72%, and the fast expansion of cities creates new opportunities and fuels economic growth but also presents many difficulties such as environmental pollution, traffic congestion, and expensive housing prices. The improvement in people's living standards has made them pay greater attention to the quest for an excellent quality of life. As the main carrier of regional innovation and economic development, cities have assumed unprecedented historical missions and contemporary responsibilities on the way to implement the new development concept and achieve sustainable development goals, and how to accelerate the construction of innovative cities, specialize urban functions, and build high-quality cities has become the focus of scholars at home and abroad [1–4]. At present, China's economy has shifted to the stage of high-quality development, facing the two-way bottleneck of "neck" technology and innovation system and mechanism reform, which puts forward objective requirements for the coordinated development of urban quality and technology innovation in the new era.

The Guangdong-Hong Kong-Macao Greater Bay Area, as an important strategic highland for China to build an international technology innovation center, has different characteristics within the region and has a unique and complex pattern of two systems, three customs zones, four core cities, and different legal systems; its regional innovation capacity, regional coordinated development level, and livable and workable urban quality construction are of importance to deepen the cooperation among Guangdong, Hong Kong, and Macao and national open development [5]. Therefore, this study selects this region as the study area to analyze the spatial structure of coupled and coordinated urban quality and technology innovation in the Greater Bay Area and its influencing factors, it is important for the country to deepen the reform of technology innovation system mechanism.

## Literature review

The study of urban quality began with John Kenneth Galbraith's studies on the quality of life in American cities [6], and steadily progressed with in-depth research on themes such as livability and sustainability [7]. As the global urbanization process intensified, scholars began to re-examine urban quality, and the multifarious challenges of urban environmental load and ecological pressure induced by economic development came to the fore [8, 9]. The meaning of urban quality has increasingly moved from a single element to a multidimensional and interconnected system, with economic, service, cultural, and ecological components all playing major roles in determining urban quality [10–12]. Many academics have investigated the mutual influence and link between technology innovation and urban quality as the most active element in regional activities [13, 14]. On the one hand, improved urban quality benefits factor circulation and concentration, providing high-quality locational space and environment for innovation activities. Feldman et al. (1999) discovered that increasing city size boosts regional rivalry and economic diversification while also encouraging urban innovation [15]. According to Battke (2016) et al., urban quality benefits knowledge accumulation and spillover, and technology innovation continues to break beyond the original foundation [16]. Meanwhile, studies from the perspectives of education level [17], innovation supply and demand [18], and policy and facility environment [19] have found that urban quality facilitates STI. On the other hand, the development of technology innovation level is the key to the transformation of urban industrial structure, the improvement of spatial utilization efficiency, the improvement of the ecological environment, and the improvement of social construction level, which injects strong dynamic energy for further optimization of urban quality. Scholars have confirmed, using model derivation and numerical simulation [20], path development study (PD) [21], and spatial Durbin model [22], that investments in technology innovation, urban governance

innovation, and regional collaborative innovation have varying degrees of positive effects on urban quality, and that technology innovation is required for sustainable urban development.

The concept of "coupling coordination" first originated in the field of physics, emphasizing the close connection and complex interactions between different subsystems [23], and in recent years, "coupling coordination" has been introduced into the analysis of the relationship between different subsystems of cities in the process of sustainable development, which is used to interpret the core meaning of sustainable development of urban systems [24], reflecting the closeness of the interaction between different urban subsystems. At present, most studies focus on considering sustainable development issues such as socio-economic and ecological environment [25], urbanization and carbon emission [26], innovation and eco-efficiency [27], and some scholars consider multi-system coupling coordination from multiple perspectives of production-life-ecology [28] or economy-society-ecology [29, 30]. These studies cover the evaluation system of urban quality, but most of them ignore the important role of human activities and their demand environment for sustainable urban development [31].

The aforementioned literature has yielded a great number of exploratory results in both theoretical and empirical aspects, but it has been difficult to lead the development of current regional development methods. The positive interaction between urban quality and technology innovation has become an important issue to be solved in the modernization of urban governance and coordinated regional development, but most of the current studies only stay at the stage of simple measurement and evaluation of the correlation [14, 22], and the construction of the urban quality evaluation system is biased toward a single dimension [11], with relatively few research results. Furthermore, the drivers of coupling coordination distribution are more focused on qualitative exploration, lacking quantitative attribution by spatial statistical methods, which easily obscures the geographical heterogeneity of the drivers and the synergistic or antagonistic effects between the factors [32].

As a result, this study will use panel data from 11 cities in the Guangdong-Hong Kong-Macao Greater Bay Area from 2011 to 2020 as the sample, measure the coupling relationship between urban quality and technology innovation using the coupling coordination model, explore the state characteristics and spatial structure of the linkage between the two using trend surface analysis and standard deviation ellipse analysis, and finally use the GeoDetector model and geographically weighted regression model(GWR) at the land level to Finally, the internal driving mechanism of the coupling coordination influence factors is thoroughly examined at the geographical probe model and geographically weighted regression model scales, in order to provide empirical support and a decision-making basis for the development of the benign interaction between urban quality and technology innovation in the Guangdong-Hong Kong-Macao Greater Bay Area.

The marginal contribution of this article is as follows: (1) There are many studies on urban quality and technology innovation now, but there are relatively few studies on the relationship between them, and they are limited to one-way linear relationships. In this paper, we analyze the interaction mechanism between urban quality and technology innovation using system theory, summarize the coupling and coordination effects of the two systems, and analyze the evolution law of the interaction between the two at the empirical level. (2) Since the statistical indicators of Hong Kong and Macao are not consistent with those of the mainland, there is no unified urban quality evaluation index system in the Guangdong-Hong Kong-Macao Greater Bay Area. Under the overall framework of the Guangdong-Hong Kong-Macao Greater Bay Area, this paper refers to the national standard "New Urbanization—Quality City Evaluation Index System" for the first time to portray the multi-attribute system characteristics of urban quality.

However, there are still some limitations to this study: At present, the article's research on the spatial structure and driving mechanism of the coupling coordination of urban quality and technology innovation is only limited to the Guangdong-Hong Kong-Macao Bay Area, and it can be extended to the national domain in the future to compare multiple city clusters.

The remainder of the paper is structured as follows. The third section provides an overview of the research design, indicator selection, and research methodology. The fourth section introduces the research topic and data sources. The fifth section highlights the findings of the geographical and temporal evolution analysis of coupling coordination. The sixth section delves into the driving forces behind coupling coordination influence factors. The seventh section examines the study's findings and makes recommendations.

## Materials and methods

### Urban quality and technology innovation coupling coordination mechanism

Throughout the history of the world's urban development, the prosperity of cities cannot be separated from the progress of technology, and the generation of technology cannot be separated from the development of cities. The two systems of urban quality and technology innovation are independent and interconnected, mutually supportive and mutually constraining, maintaining structural stability and self-organizing ability. Coupling emphasizes that the two systems influence each other through internal or external effects, while coordination reflects how closely the two systems interact and adapt to each other. From the system perspective, the coupling and coordination of urban quality and science and technology innovation refer to the benign interaction between the two systems, resulting in a synergistic effect of 1+1>2, so that the overall function is greater than the sum of the functions of the two systems operating independently. Urban quality is an important condition in support of science and technology innovation, and science and technology innovation is the main driving force of urban quality. The two interact and promote each other, and when the development of one side is hindered, it will certainly affect the development of the other side.

On the one hand, city quality is the bearing system of regional space, which provides sufficient innovation factors and a superior innovation environment for science and technology innovation, effectively reduces the institutional threshold and communication cost of innovation subjects, and provides the necessary guarantee for the realization of science and technology innovation. The improvement of city quality means the improvement of residents' income level, the increase of employment opportunities, the optimization of the industrial structure and the improvement of environmental quality, which can expand the market demand for production and living services, further gather talents, capital and other innovation factors to operate and develop together in a certain area, attract different enterprises, universities and other innovation subjects to cooperate professionally and enhance the vitality of urban innovation. In addition, the optimization of city quality is inevitably accompanied by better infrastructure and information services, which can effectively reduce the communication barriers among innovation subjects, promote the exchange and transmission of knowledge, and also provide favorable conditions for the generation of regional innovation culture. At the same time, urban quality is an excellent practice place for new knowledge and technology, and the positive feedback of improved efficiency of urban space operation is a necessary basis for testing and improving innovation results, thus promoting the enhancement of science and technology innovation.

On the other hand, technology innovation is the output system of regional space, an important driving force for the coordinated development of economic, social, demographic, and

environmental factors, and continues to optimize the spatial layout and operational efficiency of urban quality. Science and technology innovation undertakes the functions of cultivating innovative talents and researching technical knowledge, driving technology iteration and upgrading and industrial structure optimization in urban quality, and further restructuring of urban and rural areas, which can prompt the flow of innovation factors from labor-intensive industries to technology-intensive industries, thus improving labor productivity and investment yield of industries and optimizing the development mode of urban quality. At the same time, the application of innovations in healthcare and education will also promote more new industries to emerge, which promotes the quality of life of residents and the level of social services, and is conducive to the continuous improvement of urban quality.

## Evaluation index system construction

**Urban quality evaluation indexes.** A systematic and comprehensive urban quality evaluation system has yet to be developed in the Guangdong-Hong Kong-Macao Greater Bay Area due to the complicated external environment, multiple influencing variables, and regional synergy system obstruction. To some extent, to alleviate the impediment to the integrated development of urban clusters in the Guangdong-Hong Kong-Macao Greater Bay Area, it is critical to transcend the limitations of a single perspective in order to accurately and comprehensively select appropriate evaluation indicators that reflect the multi-attribute system characteristics of urban quality in the Greater Bay Area. This paper adheres to the goal of prioritizing development and quality, draws on the national standard New Urbanization-Quality City Evaluation Index System (GB/T 39497–2020), takes into account the statistical peculiarities of Hong Kong and Macao, and conducts inductive screening based on the principles of systematicity, scientificity, representativeness, and accessibility. The Guangdong-Hong Kong-Macao Greater Bay Area Comprehensive Index System of Urban Quality is constructed by selecting five core indicators and 18 auxiliary indicators, including Economic Development, Cultural Development, Ecological Environment, Public Services, and Residents' Life (Table 1).

**Technology innovation evaluation indicators.** Under the competitive landscape where digitalization and intelligent transformation are becoming increasingly prominent trends in urbanization, although the Guangdong-Hong Kong-Macao Greater Bay Area has a larger scope and higher concentration of technology-intensive manufacturing industries compared with urban agglomerations such as Beijing-Tianjin-Hebei and the Yangtze River Delta, problems such as insufficient research funding, a lack of basic and high-end talents, imperfect collaborative innovation mechanisms, and restricted core technologies still exist [33]. Based on this, this paper, based on the bottlenecks of technology innovation in the Guangdong-Hong Kong-Macao Greater Bay Area, combines the previous research results [34] with China's regional innovation capability monitoring index system and selects the three primary indicators of innovation input, innovation output, and innovation environment and the seven secondary indicators belonging to them to build a complete and reflective technology innovation index system for the Guangdong-Hong Kong-Macao Greater Bay Area, as shown in Table 2 shows.

## Data modeling

**Entropy-weighting TOPSIS model.** In the theory of information, entropy is a measure of the degree of uncertainty of the system, and the amount of information is measured by the size of entropy, which can effectively avoid the interference of human factors [35]. If the entropy value of an indicator is smaller, it indicates that it provides more information, less uncertainty, and higher weight in the objective evaluation. This paper breaks through the drawback that

**Table 1. Urban quality index system.**

| | Level 1 indicators | Level 2 indicators | Units | Indicator Properties |
|---|---|---|---|---|
| Urban Quality | Economic Development | GDP per capita | RMB | + |
| | | General public budget revenue | 10000RMB | + |
| | | Tertiary industry added value | 10000RMB | + |
| | | Urbanization rate of the resident population | % | + |
| | | Total actual utilization of foreign capital per 10 billion yuan of GDP | RMB billion | + |
| | Cultural Development | Urban registered unemployment rate | % | - |
| | | Compulsory education teacher-student ratio | % | + |
| | | Public library holdings per capita | Roll | + |
| | Ecological Environment, | Energy consumption per unit of GDP | Standard coal / 10,000 RMB | - |
| | | Annual average concentration of fine particulate matter (PM2.5) | μg/m$^3$ | - |
| | | Environmental Protection Inputs | 10,000 yuan | + |
| | Public Services | The number of public transport vehicles owned by 10,000 people in the city | Unit | + |
| | | The death rate of production safety accidents with a GDP of 100 million yuan | % | - |
| | | Economic externality | % | + |
| | Residents' Life | Number of hospital beds per 10,000 people | Sheet | + |
| | | Number of doctors per 10,000 people | Per | + |
| | | Per capita disposable income of urban residents | RMB | + |
| | | Increase in consumer price index | % | - |

the traditional TOPSIS method cannot express the relative importance of each index weight, combines the entropy weight method with the TOPSIS method, improve the method of finding the Euclidean distance between the positive ideal solution and the negative ideal solution of each city, and can make a more accurate comprehensive evaluation of the urban quality and technology innovation in the Guangdong-Hong Kong-Macao Greater Bay Area. The particular steps are as follows:

In the first step, the original data matrix is formed $X = (x_{ij})_{n \times m}$, where $x_{ij}$ signifies the value of the ith indicator in the jth city, and n and m are the numbers of evaluation indicators and evaluation cities, respectively.

$$X = \begin{bmatrix} x_{11} & x_{12} & \cdots & x_{1m} \\ x_{21} & x_{22} & \cdots & x_{2m} \\ \vdots & \vdots & \vdots & \vdots \\ x_{n1} & x_{n2} & \cdots & x_{nm} \end{bmatrix} \tag{1}$$

**Table 2. Technology innovation index system.**

| | Level 1 Indicators | Level 2 Indicators | Units | Indicator Properties |
|---|---|---|---|---|
| technology innovation | Innovation inputs | Internal expenditure on R&D expenses | 10,000 RMB | + |
| | | Number of R&D researchers | Per | + |
| | Innovative Output | Number of patents granted | Unit | + |
| | | Number of publications | Unit | + |
| | Innovation Environment | Financial Science and Technology Grant | Per | + |
| | | Information Infrastructure | Per | + |
| | | Number of students enrolled in general higher education institutions | Per | + |

In the second step, the positive indicators in the matrix are standardized and normalized, respectively, by the following formulae:

$$a_{ij} = \frac{x_{ij} - x_{i\min}}{x_{i\max} - x_{i\min}} \tag{2}$$

$$b_{ij} = \frac{a_{ij}}{\sum\limits_{j=1}^{11} a_{ij}} \, (i = 1, 2, \cdots, 16) \tag{3}$$

In the third step, measure the entropy value of each indicator.

$e_i = -\frac{1}{\ln 11} \sum\limits_{j=1}^{11} b_{ij} * \ln b_{ij} (i = 1, 2, \cdots, 16)$, and there is the assumption when $b_{ij} = 0$,

$b_{ij} * \ln b_{ij} = 0$. The entropy weights of indicator i can be obtained.

$$w_i = \frac{1 - e_i}{\sum\limits_{i=1}^{16} 1 - e_i} \, (i = 1, 2, \cdots, 16) \tag{4}$$

In the fourth step, the maximum and minimum values of each indicator in all cities are determined using the matrix $A = (a_{ij})_{n \times m}$ obtained by normalization in the second step, and positive and negative ideal solution vectors are constructed respectively.

$$\begin{aligned} A^+ &= (\max\{a_{11}, a_{12}, \cdots a_{1m}\}, \max\{a_{21}, a_{22}, \cdots a_{2m}\}, \cdots, \max\{a_{n1}, a_{n2}, \cdots a_{nm}\}) \\ &= (A_1^+, A_2^+, \cdots, A_n^+) \end{aligned} \tag{5}$$

$$\begin{aligned} A^- &= (\min\{a_{11}, a_{12}, \cdots a_{1m}\}, \min\{a_{21}, a_{22}, \cdots a_{2m}\}, \cdots, \min\{a_{n1}, a_{n2}, \cdots a_{nm}\}) \\ &= (A_1^-, A_2^-, \cdots, A_n^-) \end{aligned} \tag{6}$$

In the fifth step, the weighted Euclidean distances of the positive and negative ideal solutions for each city are determined independently by combining the entropy weights obtained in the third step.

$$D_j^+ = \sqrt{\sum\limits_{i=1}^{n} w_i (a_{ij} - a_i^+)^2} \, (j = 1, 2, \cdots, m) \tag{7}$$

$$D_j^- = \sqrt{\sum\limits_{i=1}^{n} w_i (a_{ij} - a_i^-)^2} \, (j = 1, 2, \cdots, m) \tag{8}$$

In the sixth step, find the proximity of each city to the optimal solution.

$$U_j = \frac{D_j^-}{D_j^+ + D_j^-}, \, 0 \leq U_j \leq 1 \tag{9}$$

The closer the value of $U_j$ when is to 1, the closer the city's urban quality (STI) score is to the ideal value.

**Coupling coordination degree model.** This paper introduces the coupling coordination degree model to quantify the interaction and mutual influence between the two systems of urban quality and technology innovation, to break through the limitation that a single

coupling model cannot accurately reflect the development level of each of the two systems and more accurately assess the degree of their coordinated development. For the model of the dual system of urban quality ($U_1$) and technology innovation ($U_2$) in this research, the processes for determining the coupling coordination degree are as follows.

In the first step, the coupling degree which reflects the magnitude of the correlation between the two systems is calculated:

$$C = 2\sqrt{\frac{U_1 U_2}{(U_1 + U_2)^2}},\ 0 \leq C \leq 1 \tag{10}$$

The higher the coupling degree, the more the city focuses on the coupling development of innovation quality and innovation atmosphere; the lower the coupling degree, the less healthy the development relationship between the two systems.

In the second step, calculate the composite reconciliation index reflecting the high level of coordination between the two systems.

$$T = aU_1 + bU_2 \tag{11}$$

Where $a+b = 1$, $a$ and $b$ is coefficient to be determined, describing the relative importance of the two systems of urban quality and technology innovation. In this paper, the relative importance of the two systems of urban quality and technology innovation is considered to be the same, so this paper makes $a = b = 0.5$.

The third step is to find the coupling coordination.

$$D = \sqrt{C \times T} \tag{12}$$

Based on the data distribution and with reference to previous research results [34], this paper divided the coupling coordination into seven levels: $0.0 < D \leq 0.2$ for severe disorder; $0.2 < D \leq 0.3$ for moderate disorder; $0.3 < D \leq 0.4$ for mild disorder; $0.4 < D \leq 0.6$ for barely coordination; $0.6 < D \leq 0.7$ for primary coordination; $0.7 < D \leq 0.9$ for well coordination; $0.9 < D \leq 1.0$ is high quality coordination.

## Trend surface analysis

The trend surface analysis is based on the principle of regression analysis, the binary nonlinear function is fitted by the least squares approach, and a smooth mathematical surface is utilized to represent the general trend and distribution pattern of geographic elements in space [36]. Trend surface is analyzed by applying the least squares (OLS) method to fit a two-dimensional nonlinear function through the principle of regression analysis, the core of which is to derive the trend surface from the actual observations so that the residual sum of squares tends to be minimized. The spatial distribution pattern of geographic elements is simulated to show the variation of geographic elements in geographical space. In the actual spatial trend surface simulation, any function can be approximated by polynomials in an appropriate range, and the number of polynomials is adjusted according to the spatial situation of things so that the regression equation is suitable for the needs of the actual problem. The formula is as follows:

$$P_j(x_j, y_j) = Z_j(x_j, y_j) + \varepsilon \tag{13}$$

$$Z_j(x_j, y_j) = \beta_0 + \beta_1 x + \beta_2 y + \beta_3 x^2 + \beta_4 y^2 + \beta_5 xy \tag{14}$$

Where $(x_j, y_j)$ is the planar spatial coordinate of city j and $Z_j(x_j, y_j)$ is the trend surface function. The parameters $\beta_0, \beta_1, \beta_2, \beta_3, \beta_4, \beta_5$ are determined based on the observations $z_j, x_j, y_j$ to minimize the residual sum of squares.

**The gravity center (GC) and standard deviation ellipse (SDE) method.** The gravity center (GC) and standard deviation ellipse (SDE) method is a spatial statistical method that can quantitatively analyze the multidimensional spatial characteristics of an attribute in the region. In this paper, we use the GC model and SDE method to analyze the spatial wholeness of the geographic element characteristics of the coupling coordination of urban quality and technology innovation in the Guangdong-Hong Kong-Macao Greater Bay Area, and reflect the transmutation pattern of the overall spatial pattern by the direction of movement and distance of movement of the gravity center of the coupling coordination degree and the change characteristics of the ellipse declination, area, long semi-axis and short semi-axis, with the following equation [37].

$$X = \sum_{i=1}^{n} D_i x_i / \sum_{i=1}^{n} D_i \tag{15}$$

$$Y = \sum_{i=1}^{n} D_i y_i / \sum_{i}^{n} D_i \tag{16}$$

$$tan\theta = \frac{(\sum_{i=1}^{n} \tilde{x}_i^{\,2} - \sum_{i=1}^{n} \tilde{y}_i^{\,2}) + \sqrt{(\sum_{i=1}^{n} \tilde{x}_i^{\,2} - \sum_{i=1}^{n} \tilde{y}_i^{\,2}) + 4(\sum_{i=1}^{n} \tilde{x}_i \tilde{y}_i)^2}}{2 \sum_{i=1}^{n} \tilde{x}_i \tilde{y}_i} \tag{17}$$

$$\delta_x = \sqrt{\sum_{i=1}^{n} (\tilde{x}_i cos\theta - \tilde{y}_i sin\theta)^2 / n} \tag{18}$$

$$\delta_y = \sqrt{\sum_{i=1}^{n} (\tilde{x}_i sin\theta - \tilde{y}_i cos\theta)^2 / n} \tag{19}$$

Where $X$ and $Y$ are the coordinates of the gravity center of the coupling coordination of urban quality and technology innovation, $x_i$ and $y_i$ are the geographic coordinates of the cities, the ellipse azimuth $\theta$ is the angle formed by clockwise rotation in the due north direction to the long axis of the ellipse, $\tilde{x}_i$ and $\tilde{y}_i$ are the deviations of the coordinates from the geographic coordinates of each city to the gravity center, $\delta_x$ and $\delta_y$ are the standard deviations along the x-axis and y-axis, respectively.

**The geodetector model.** The geodetector model makes up for the shortcomings of the lack of spatial differentiation studies and the limitations of assumption conditions in traditional statistics while combining GIS spatial overlay technology and set theory to effectively identify synergistic or antagonistic relationships among multiple factors. In this paper, we utilize a geographic detector model to examine the determinants and methods of influence of diverse factors and their interaction terms on the coupling coordination of urban quality and technology innovation in the Guangdong-Hong Kong-Macao Greater Bay Area. The formula

is as follows [38]:

$$q = 1 - \frac{\sum_{h=1}^{L} N_h \delta_h^2}{N \delta_h^2} \tag{20}$$

$$\lambda = \frac{1}{\delta^2} \left[ \sum_{h=1}^{L} \bar{Y}_h^2 - \frac{1}{N} \left( \sum_{h=1}^{L} \sqrt{N_h} \bar{Y}_h \right)^2 \right] \tag{21}$$

Where $N_h$ and $\delta_h$ are the unit value and variance of layer h, respectively, and the q-value reflects the degree of explanation of the factor on the coupling coordination, and $\lambda$ can test the significance of the q-value.

**Geographically weighted regression (GWR).** The Geographically Weighted Regression (GWR) model is a modified local spatial linear regression model proposed based on the idea of local smoothness, which embeds the attribute data in spatial locations and reflects the influence of the explanatory variables on the explained variables with changes in spatial locations [39], with the following equation.

$$Y_i = \pi_0(u_i, v_i) + \sum_k \pi_k(u_i, v_i) X_{ik} + \varepsilon_i \tag{22}$$

Where $(u_i, v_i)$ is the coordinates of the spatial geographic location of the ith sample point.

## Study area and data sources

**Overview of the study area.** The Guangdong-Hong Kong-Macao Greater Bay Area is located in the coastal region of South China (111°21′~114°53′E, 21°28′~24°29′N), and consists of the two special administrative regions of Hong Kong and Macau and the nine Pearl River Delta(PRD) cities of Guangzhou, Shenzhen, Zhuhai, Foshan, Zhongshan, Jiangmen, Huizhou, Dongguan, and Zhaoqing in Guangdong Province. The Guangdong-Hong Kong-Macao Greater Bay Area has a leading level of economic development, a complete industrial system, obvious cluster advantages, and the special advantages of "one country, two systems, and three customs zones," and has always been one of the regions with the highest degree of openness and economic vitality in China. In February 2019, the Central Committee of the Communist Party of China and the State Council issued "the outline of the development plan of the Guangdong-Hong Kong-Macao Greater Bay Area," making it clear that the Guangdong-Hong Kong-Macao Greater Bay Area will not only become a vibrant world-class city cluster, an international center of technology innovation, an important support for the construction of the Mainland, Hong Kong, and Macao in-depth cooperation demonstration area, but also a quality living circle that is pleasant to live, work, and travel, and become a model of high-quality development.

**Sample selection and data sources.** To investigate the coupling coordination mechanism and spatial and temporal evolution characteristics of urban quality and technology innovation in a holistic framework, this paper takes 11 cities in the Guangdong-Hong Kong-Macao Greater Bay Area as the research objects, and the study spans the period 2011–2020. The data used in the study were obtained from the Yearbook of Science and Technology of China, the Statistical Bulletin of the Guangdong Bureau of Science and Technology Statistics, the Hong Kong Statistics Department, the Statistics and Census Service of Macao, the World Bank, the Web of Science Core Collection database, and the State Intellectual Property Office.

In terms of the treatment of indicators, on the value added of tertiary industries, the nine PRD cities selected the relevant data on the value added of tertiary industries from the

**Table 3. Urban quality index descriptive statistics.**

| Variable | Obs | Mean | Std. Dev. | Min | Max |
|---|---|---|---|---|---|
| GDP per capita | 110 | 164220.49 | 344297.77 | 29683.882 | 3515051 |
| General public budget revenue | 110 | 20081.173 | 44529.541 | 499.778 | 199318.66 |
| Tertiary industry added value | 110 | .562 | .194 | .357 | 1.128 |
| Urbanization rate of the resident population | 110 | 34.573 | 27.849 | 1 | 84 |
| Total actual utilization of foreign capital per 10 billion yuan of GDP | 110 | 1.718 | 3.164 | .092 | 21.195 |
| Urban registered unemployment rate | 110 | 2.38 | .511 | 1.2 | 5.8 |
| Compulsory education teacher-student ratio | 110 | 2.38 | .511 | 1.2 | 5.8 |
| Public library holdings per capita | 110 | 1.094 | .716 | .055 | 2.792 |
| Energy consumption per unit of GDP | 110 | 1950.398 | 1615.899 | 97.804 | 6294.2 |
| Annual average concentration of fine particulate matter (PM2.5) | 110 | 36.128 | 12.131 | 15.7 | 66 |
| Environmental Protection Inputs | 110 | 32.175 | 53.254 | 0 | 331.635 |
| The number of public transport vehicles owned by 10,000 people in the city | 110 | 9.498 | 5.631 | 1.183 | 21.771 |
| The death rate of production safety accidents with a GDP of 100 million yuan | 110 | .064 | .05 | .009 | .22 |
| Economic externality | 110 | .207 | .166 | .005 | .596 |
| Number of hospital beds per 10,000 people | 110 | 33.627 | 10.145 | 15.146 | 56.783 |
| Number of doctors per 10,000 people | 110 | 22.772 | 5.778 | 13.065 | 46.002 |
| Per capita disposable income of urban residents | 110 | 100765.26 | 135387.17 | 19039.65 | 505875.63 |
| Increase in consumer price index | 110 | 1.392 | 1.413 | 0 | 6.11 |

statistical yearbook. According to the yearbook of the Hong Kong region, we selected the value-added industries except for manufacturing and construction as the value-added of the Hong Kong tertiary industries. In Macau, the value added of industries other than manufacturing, water and gas production and supply, and construction is selected as the value-added of tertiary industries according to the yearbook. internal expenditure on R&D funds measures the level of scientific research investment in a region, and the data of the nine PRD cities are obtained from the statistical yearbook, while the data of Hong Kong and Macau are obtained from the R&D expenditure indicators of the World Bank. Meanwhile, the relevant amount of data involving the currency amounts of Hong Kong and Macau concerning currency representation in this article is converted according to the relevant rate of the current year's exchange rate (Tables 3 and 4).

**Technology path.** In this paper, we first use the entropy-weighted TOPSIS method, the coupled coordination model and the centre of gravity-standard deviation ellipse method to deeply reveal the spatio-temporal evolutionary characteristics of the coupled coordination of urban quality and science, technology and innovation in Guangdong, Hong Kong and Macao Greater Bay Area, with the main focus on the dynamic law of the temporal evolutionary trend

**Table 4. Technology innovation descriptive statistics.**

| Variable | Obs | Mean | Std. Dev. | Min | Max |
|---|---|---|---|---|---|
| Internal expenditure on R&D expenses | 110 | 1966509.1 | 2783581 | 10596.662 | 15108100 |
| Number of R&D researchers | 110 | 53226.411 | 65145.153 | 711 | 345780 |
| Number of patents granted | 110 | 7498.073 | 12354.993 | 165 | 57050 |
| Number of publications | 110 | 54.7 | 31.621 | 1 | 109 |
| Financial Science and Technology Grant | 110 | 939.49 | 809.463 | 45.93 | 2988.86 |
| Information Infrastructure | 110 | 511381.25 | 987137.92 | 6000.161 | 5549818 |
| Number of students enrolled in general higher education institutions | 110 | 26106.064 | 36974.226 | 19 | 222412 |

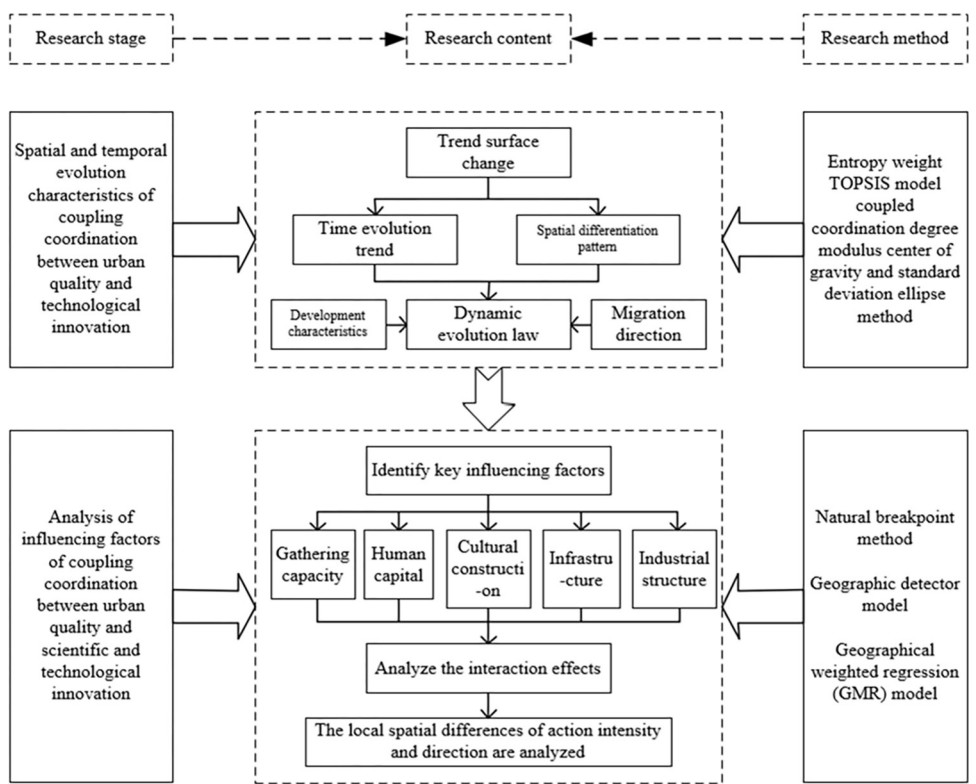

**Fig 1. Technology roadmap of the research.**

and the spatial differentiation pattern. Secondly, using the geodetector model and geographically weighted regression (GMR) model, the interactions between the main influencing factors and factors, as well as the spatial variability of the strength and direction of the factors' effects on the coupled coordination are further analysed. This technology Path provides an effective analytical framework for an in-depth understanding of the coupled coordination of urban quality and science and technology innovation in the Guangdong-Hong Kong-Macao Greater Bay Area and its influencing factors, as shown in Fig 1.

## Results and discussion

### Spatio-temporal evolution characteristics of the coupling coordination of urban quality and technology innovation

**Trend surface analysis results.** The entropy-weighted TOPSIS method and the coupling coordination model are used to calculate the coupling coordination model degree of urban quality and technology innovation in the Guangdong-Hong Kong-Macao Greater Bay Area from 2011 to 2020, and the trend surface analysis tool in ArcGIS software is used to spatially visualize and express it through a smooth mathematical surface, as shown in Fig 2.

On the whole, the coupling coordination of the Guangdong-Hong Kong-Macao Greater Bay Area from 2011 to 2020 shows a spatial distribution pattern of "high in the middle and low around," with a logarithmic curve shape of "high in the east and low in the west" in the east-west direction, and an inverted U-shaped parabolic shape in the north-south direction, where the slope and curvature are gradually increasing.

In terms of the temporal evolution of the trend surface, the curve characteristics in the east-west direction have remained relatively stable over the last ten years, and the coupling and

                                              

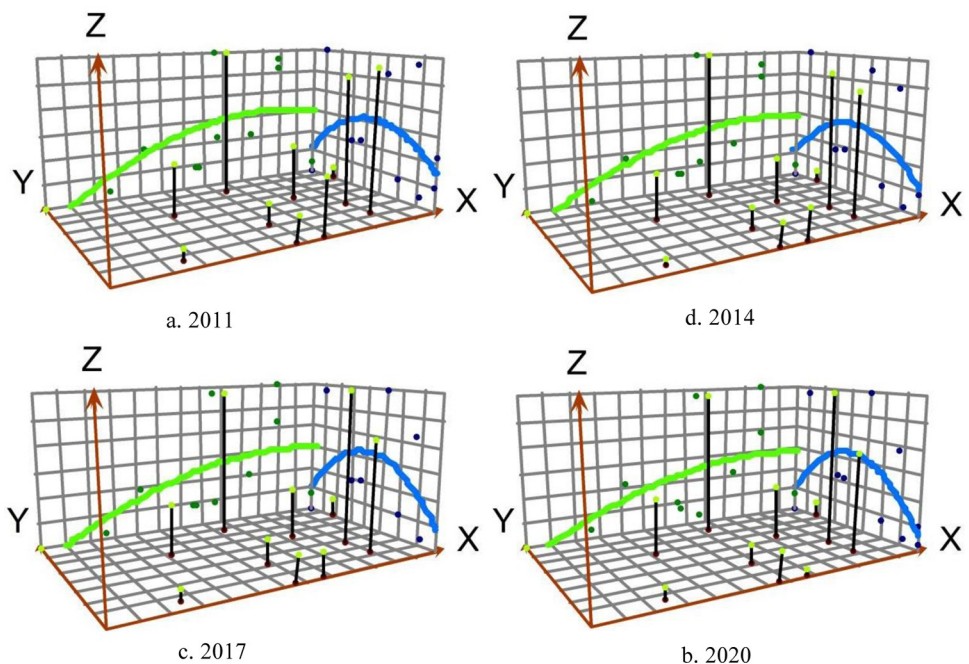

**Fig 2. Trend surface changes in the coupling coordination of urban quality and technological innovation in the Guangdong-Hong Kong-Macao Greater Bay Area.**

coordination level of cities on the Pearl River's west bank, such as Zhaoqing, Jiangmen, and Zhongshan, is relatively weak and has yet to produce any significant breakthrough. At the same time, the trend surface always maintains the evolution pattern of convexity towards the middle, indicating that the spatial polarization characteristics of the coupled and coordinated development of urban quality and technological vation in the Greater Bay Area are obvious and the level of benign interaction of cities located along the central part is becoming increasingly prominent. In line with the findings of Hechang Cai et al. [39] on technology transfer in the Guangdong-Hong Kong-Macao Greater Bay Area, the coupling and coordination development is also manifested in the significant "strong is always strong, the weak is always weak" Matthew effect.

## Spatial pattern evolution of coupling coordination degree

Based on the study of the overall geographical evolution trend, the spatial distribution characteristics of the type of coupling coordination of urban quality and technological innovation in the Guangdong-Hong Kong-Macao Greater Bay Area are analysed, as shown in Table 5.

The coupling coordination degree of urban quality and technology innovation in the Guangdong-Hong Kong-Macao Greater Bay Area shows an upward trend from 2011 to 2020, with most cities leaping toward a higher level, but individual cities such as Zhaoqing and Macao also experiencing a fallback. The value of the coupling coordination degree increased from 0.2322–0.5308 in 2011 to 0.2203–0.7298 in 2020, indicating that the overall coordination between urban quality and technological innovation is increasing and the coupling relationship is getting closer, but the low-value area still fails to On the whole, The type of coupling coordination is predominantly moderate dissonance and mild dissonance, and the Guangdong-Hong Kong-Macao Greater Bay Area has established a spatial divergence pattern with the high coupling coordination area stretching in all directions with Guangzhou-Shenzhen as the core.

Table 5. Spatial and temporal divergence of coupling coordination in the Guangdong-Hong Kong-Macao Greater Bay Area.

| City | Coupling Coordination Degree | | | |
|---|---|---|---|---|
| | **2011** | **2014** | **2017** | **2020** |
| HongKong | Barely Coordination | Barely Coordination | Barely Coordination | Barely Coordination |
| Macao | Moderate disorder | Moderate disorder | Moderate disorder | Moderate disorder |
| Guangzhou | Barely Coordination | Barely Coordination | Primary Coordination | Well Coordination |
| Shenzhen | Barely Coordination | Barely Coordination | Primary Coordination | Well Coordination |
| Zhuhai | Moderate disorder | Moderate disorder | Moderate disorder | Mild disorder |
| Foshan | Mild disorder | Mild disorder | Mild disorder | Barely Coordination |
| Huizhou | Moderate disorder | Moderate disorder | Moderate disorder | Moderate disorder |
| Dongguang | Mild disorder | Mild disorder | Mild disorder | Barely Coordination |
| Zhongshan | Moderate disorder | Moderate disorder | Moderate disorder | Moderate disorder |
| Jiangmen | Moderate disorder | Moderate disorder | Moderate disorder | Moderate disorder |
| Zhaoqing | Moderate disorder | Severe disorder | Moderate disorder | Moderate disorder |

Specifically, from 2011 to 2014, the coordination level of urban technology innovation was relatively stable, the development was slightly slow, the spatial location heterogeneity was significant, the cities with moderate disorder were scattered at the edge of the Greater Bay Area, and only Guangzhou, Shenzhen, and Hong Kong reached the barely coordinated level. 2014–2017, the coupling coordination level of Guangzhou and Shenzhen rises to the major coordination level. The "dual-core connectivity" improves the coordinated growth of nearby cities. Guangzhou and Shenzhen have relatively better urban quality environments and high technology innovation strengths, and the systems display an orderly spiral optimization of the growing state, which may give full play to the role of growth poles to radiate the nearby regions [40]. 2017–2020, the sequential adoption of a number of Guangdong-Hong Kong-Macao Greater Bay Area policy measures has provided strong policy dividends for the coordinated resonance of urban quality and technology innovation, the interaction and interchange, and open cooperation of the cities develop closer and closer, and the coupling coordination level of urban quality and technology innovation is rapidly improved. Guangzhou and Shenzhen have achieved the stage of well coordination, whereas Foshan and Dongguan have entered the stage of barely cooperation. However, the coupling coordination level of edge cities such as Zhaoqing, Huizhou, and Jiangmen has not yet changed dramatically. On the one hand, the edge cities have not yet established stable cooperation links with the core cities and lack suitable ways to leapfrog; on the other hand, the edge cities have a single industrial structure and show poorer risk tolerance in the face of external shocks from the trade war between China and the United States and the new crown epidemic in the past two years, making the coupling coordination development more vulnerable to external disturbances, break through the dilemma.

## The gravity center (GC) and standard deviation ellipse (SDE) analysis

This paper employs the GC-SDE model of ArcGIS to reveal the development characteristics and migration direction of coupling coordination from 2011 to 2020 in order to grasp the dynamic evolution law of coupling coordination of urban quality and technology innovation in the Guangdong-Hong Kong-Macao Bay Area as a whole, as shown in Table 6.The movement of the gravity center of coupling coordination is shown in Table 7.

From the gravity center migration trajectory, the gravity center of coupling coordination in the Guangdong-Hong Kong-Macao Greater Bay Area from 2011 to 2020 is all located within the administrative boundaries of Guangzhou City, adjacent to the border of Dongguan City, with an overall migration trend to the northeast. Guangzhou is the coordination center of

**Table 6. Standard deviation ellipse.** (Unit: degrees, kilometers, square kilometers).

| Year | Long semi-axis | Short semi-axis | Rotation | Area |
|---|---|---|---|---|
| 2011 | 8.4205 | 7.0831 | 102.9086 | 18735.38 |
| 2014 | 8.2571 | 6.9578 | 105.2373 | 18047.99 |
| 2017 | 8.1898 | 6.9175 | 101.4377 | 17797.08 |
| 2020 | 8.1346 | 6.8515 | 100.8367 | 17508.54 |

urban quality and technology innovation in the Guangdong-Hong Kong-Macao Greater Bay Area. From the relative positions of the gravity center of coupling coordination and the mass center, the gravity center is all located in the southeast direction of the mass center, indicating that the level of coupling coordination development in the southeast direction of the Guangdong-Hong Kong-Macao Greater Bay Area is better than other regions, and Shenzhen and Hong Kong have outstanding advantages in urban quality and technology innovation, forming a strong innovation synergy through cluster collaboration.

From the movement of the gravity center of coupling coordination (Table 6), the distance of the gravity center of coupling coordination in the northeast direction of the Greater Bay Area from 2011 to 2020 is 2.888 km, 1.027 km, and 1.380 km. The improvement of urban coupling coordination in the northeast region is greater than that in other regions of the Greater Bay Area and the unbalanced development trend of coupling coordination in the north-south direction is gradually easing. Dongguan is linked to Guangzhou in the north and Shenzhen in the south, is adjacent to Hong Kong and Macao, and is in the middle of the Guangzhou-Shenzhen-Hong Kong economic corridor. It is making every effort to build a "Bay Area City, Quality Dongguan," with a high urban quality connotation, to pull the transformation and development, focusing on the "technology innovation + advanced manufacturing" urban characteristics, and greatly improving the urban landscape by realizing technology innovation and advanced manufacturing. To enhance the urban landscape by achieving deep integration and rapid development of technology innovation and urban quality. In contrast, while cities in the Bay Area's southwest have improved their level of technology innovation as a result of the Guangdong-Shenzhen-Hong Kong-Macao Tchnological Innovation Corridor's innovation radiation, a lack of high-quality public service resources, inadequate transportation infrastructure, and small-scale industries have become bottlenecks that limit the coordination and development of urban quality and science and technology.

In terms of the distribution range of the ellipse, the azimuth angle experiences a process of increasing and then decreasing from 2011 to 2020, from 102.9086 in 2011 to 100.8367 in 2020. The overall trend of counterclockwise rotation is shown by the change in the standard deviation ellipse. The standard deviation of the main axis of the ellipse, the standard deviation of the secondary axis, and the area decrease year by year, showing a contraction away from the southwest direction, indicating that the spatial divergence characteristics of the coupling and coordination of urban quality and technology innovation in Guangdong, Hong Kong, and Macao are significant, and the coupling and coordination development level of the inner region of the ellipse is significantly higher than that of the outer region, showing the development characteristics of central polarization, and the southwest direction of the Greater Bay Area is facing the risk of marginalization and isolation.

## Analysis of factors influencing the coupling coordination of urban quality and technology innovation

**Factor detection analysis.** There are obvious spatial differences in the coupling coordination of urban quality and technological innovation in the Guangdong-Hong Kong-Macao

**Table 7. Coupling coordination center of gravity shift.**

| Year | Longitude | Latitude | Orientation | Distance (km) |
|---|---|---|---|---|
| 2011 | 113.576 | 22.750 | - | - |
| 2014 | 113.599 | 22.765 | Northeast | 2.888 |
| 2017 | 113.607 | 22.770 | Northeast | 1.026 |
| 2020 | 113.610 | 22.782 | Northeast | 1.380 |
| The gravity center | 113.226 | 23.053 | - | - |

Greater Bay Area, and the identification of their influencing factors is the key to exploring the intrinsic mechanism of high-quality coordination. In this paper, concerning relevant research results [41–45], we comprehensively consider the coupling relationship between urban quality and technology innovation in the Greater Bay Area and the correlation of indicators inside and outside the system based on the availability of data in the Greater Bay Area, we assume that agglomeration capacity (unit area of the resident population), human capital (number of R&D researchers per 10,000 people), cultural development (public library collection per capita), infrastructure (road density), and industrial structure (share of tertiary industry in GDP) will have significant effects on the coupling coordination relationship between urban quality and technological innovation in the Guangdong-Hong Kong-Macao Greater Bay Area, and analyze and verify their specific influence mechanisms. The natural discontinuity method is a statistical approach that classifies according to the law of numerical statistical distribution, which may group the comparable values correctly and make the difference between the groups maximized. In this paper, according to the natural discontinuity method. The five influencing factors are divided into five levels using ArcGIS and introduced into the geographic probe for calculation, and the results of factor detection are displayed in Table 8.

Overall, agglomeration capacity, human capital, cultural development, and infrastructure pass the significance test at the 0.01 level and can significantly drive the coupling coordination development of urban quality and technology innovation, but there are significant differences in the form and acuity of the effect of each factor on the coupling coordination degree, cultural development shows better explanatory power. Among them, cultural development has the highest q value of 0.7261. It shows that cultural development can explain the coupling coordination of the Guangdong-Hong Kong-Macao Greater Bay Area to a degree of 72.61%. The city is a spatial carrier for cultivating and spreading culture. On the one hand, cultural development strengthens citizens' cultural legacies and urban cultural identity, enriches their spiritual civilization demands, and enables the quality and high-quality development of the city. On the other hand, cultural development is the basic element guarantee for scientific research, which promotes the dissemination of explicit knowledge, while positive scientific culture increases the social recognition of technology innovation, enhances the public's willingness to participate in innovation activities, and has the internal vitality to promote the development of technology innovation.

**Table 8. Factor detection of urban quality and technology innovation coupling coordination.**

| Impact factors | Value of q statistic | p-value |
|---|---|---|
| **agglomeration capacity X1** | 0.6102 | 0.0000 |
| **human capital X2** | 0.3001 | 0.0000 |
| **cultural development X3** | 0.7261 | 0.0000 |
| **infrastructure X4** | 0.6317 | 0.0000 |
| **industrial structure X5** | 0.0520 | 0.2714 |

The q-values of infrastructure, as well as agglomeration capacity, are 0.6317 and 0.6102, respectively, ranking second and third in explanatory power for coupling coordination. Infrastructure is the cornerstone of urban function realization and the material medium of spatial connection, and kilometer density is related to the convenience and smoothness of science and technology personnel interaction and innovation resource transportation in the Guangdong-Hong Kong-Macao Greater Bay Area, which provides a strong guarantee for the improvement of urban quality and technology innovation. Population agglomeration provides labor and development demand for the improvement of urban quality technological and shows a positive promotion effect on the coupling relationship. However, the industrial structure does not have a significant driving effect on the coupling coordination development of urban quality and technology innovation, indicating that the proportion of tertiary industry cannot be increased regardless of the situation, and the separate emphasis on the industrial structure may squeeze the play of other factors.

In terms of the human capital factor, it has a q value of 0.3001, the strong economic and technological vitality of the Guangdong-Hong Kong-Macao Greater Bay Area, coupled with the continuous force of the talent introduction policy over the years, has attracted a large number of high-quality talents at home and abroad, gathering high-quality and sufficient human resources and showing some positive promotion used for coupling coordination. Due to the large scale and stability of regional talent, its positive effect on the coupling relationship between urban quality and technology innovation is relatively weak compared to the obvious growth rate of other influencing factors.

**Interaction detection analysis.** The GeoDetector can detect the genuine interactions between components, including the strength, direction, and linearity of the interactions, and is not confined to the multiplicative interactions pre-specified by econometrics [37]. Based on the factor detection model estimation, four significantly influencing factors, namely agglomeration capacity, human capital, cultural development, and infrastructure, were screened for interaction detection analysis to identify the impact of interactions between different factors on coupled and coordinated development, and the results are shown in Table 9.

The impact of any two influencing factors after an interaction is greater than the impact level of individual factors, indicating that the coupling coordination of urban quality and technology innovation is the result of multiple factors acting together, indicating that a benign composite regional coordinated development mechanism has been formed in the Guangdong-Hong Kong-Macao Greater Bay Area. The interaction values of cultural development and other terms are higher than 0.80, which can effectively stimulate other factors to release their influence potential.

Guangdong, Hong Kong, and Macao are close in geographical proximity and cultural lineage, and since the "promulgation of the Guangdong-Hong Kong-Macao Greater Bay Area Cultural and Tourism Development Plan" in 2020, the governments of each region have deeply linked Lingnan cultural resources, cultivated the humanistic spirit of the Bay Area, and

**Table 9. Interaction detection of urban quality and technology innovation coupling coordination.**

| one-by-one interaction | Interaction value | Contrast value | Interaction results |
|---|---|---|---|
| X1∩X2 | 0.6620 | >max[q(X1),q(X2)] | Two Factor Enhancement |
| X1∩X3 | 0.9071 | >max[q(X1),q(X3)] | Two Factor Enhancement |
| X1∩X4 | 0.6623 | >max[q(X1),q(X4)] | Two Factor Enhancement |
| X2∩X3 | 0.8454 | >max[q(X2),q(X3)] | Two Factor Enhancement |
| X2∩X4 | 0.7259 | >max[q(X2),q(X4)] | Two Factor Enhancement |
| X3∩X4 | 0.8955 | >max[q(X3),q(X4)] | Two Factor Enhancement |

relied on developed information technology to drive the synergistic development of multiple industries based on cultural integration hubs. Among them, the greatest interaction value of 0.9071 was identified for cultural development and agglomeration capacity, demonstrating that strong cultural public service capability is also a major driving force for talent attraction and agglomeration.

It is worth mentioning that, compared with the single-factor detection results, the explanatory power of human capital on the coupling coordination degree of urban quality and technology innovation is greatly increased after interacting with other factors. The increase of human capital will promote the flow of production factors from labor-intensive industries to knowledge-intensive and capital-intensive industries and promote the accelerated integration of capital, technology, information, culture, and other locational factors, which is conducive to technology innovation and upgrading of the overall industry, the optimal allocation of factors, and the optimization of functional space, and thus can better promote the role of multiple factors and drive the benign resonance of urban quality and technology innovation. It can be seen that human capital is a crucial aspect of the Guangdong-Hong Kong-Macao Greater Bay Area to construct a global innovation strategy highland. Governments at all levels should actively execute the "talent first" approach, strengthen the talent introduction and cultivation systems, and construct a high-quality talent pipeline.

The competition for technology innovation is essentially the competition of talents. In recent years, the Guangdong-Hong Kong-Macao Greater Bay Area has made continuous efforts to build a high-level talent highland, and the talent siphon effect is remarkable, which will provide strong support for the coordinated development of future urban quality and innovation development. At the same time, the talent policy of the Greater Bay Area, the attractiveness of science and innovation talent more and more enhanced, and the talent siphon effect more and more obvious at the same time, should work on the structure of talent, in order to promote the efficient integration of human capital and the factors, to promote scientific and technological progress while accelerating the transformation of scientific and technological achievements.

**Spatial heterogeneity analysis of impact factors based on GWR.** We further introduce the GWR model to deeply analyze the local spatial variability of the strength and direction of the coupling coordination effect of each factor on urban quality and technology innovation. The cross-sectional data of 2020 minus the corresponding indicator values of 2017 are selected and standardized, and the obtained changes in each driver are taken as the model-independent variables, the changes in the coupling coordination degree are taken as the dependent variables, and the commonly used adaptive kernel types and AICC bandwidths are selected.

Compared with the GWR regression model, the least squares analysis (OLS)—linear regression model is a traditional global regression model, which assumes that the regression parameters are independent of the spatial location of the sample, i.e., the relationship between the variables is "anisotropic" and does not change with the change of spatial location. The regression coefficients obtained are some kind of "average value" in the whole study area and cannot reflect the real spatial characteristics of the regression parameters. This assumption violates the law of spatial heterogeneity or non-smoothness in the real geographic world and cannot reflect the spatial heterogeneity between dependent variables and their influencing factors in geographic phenomena. Therefore, based on the characteristics of this study, the OLS regression results are not representative. The local regression model GWR, which applies to the geographic phenomenon of "spatial non-stationarity", is selected to establish the local regression equation at each point in the spatial range and analyze the cross-sectional data at a certain time point, which takes into account the local effects of spatial objects, has higher accuracy, and is more suitable for this study. It is more suitable for this study. It was tested that the

**Table 10. Spatial distribution of the estimated regression coefficients of the GWR model for the coupled coordination level of urban quality and technology innovation.**

| City | Agglomeration capability | Human captical | Cultural development | Infrastructure |
|---|---|---|---|---|
| HongKong | 0.0664~0.0813 | 0.4537~0.5138 | 0.8597~0.9183 | 0.4596~0.7605 |
| Macao | 0.0179~0.0664 | 0.5138~0.5620 | 0.6829~0.8597 | 0.7605~1.0839 |
| Guangzhou | 0.0931~0.1007 | 0.5620~0.6397 | 0.9183~1.277 | 0.0815~0.2943 |
| Shenzhen | 0.0664~0.0813 | 0.4537~0.5138 | 0.8597~0.9183 | 0.4596~0.7605 |
| Zhuhai | 0.0179~0.0664 | 0.5138~0.5620 | 0.6829~0.8597 | 0.7605~1.0839 |
| Foshan | -0.0057~0.0179 | 0.2773~0.4537 | 0.4507~0.6829 | 1.0839~1.3574 |
| Huizhou | 0.0813~0.0931 | 0.2773~0.4537 | 0.8597~0.9183 | 0.2943~0.4596 |
| Dongguang | 0.0813~0.0931 | 0.4537~0.5138 | 0.8597~0.9183 | 0.2943~0.4596 |
| Zhongshan | 0.0813~0.0931 | 0.5138~0.5620 | 0.6829~0.8597 | 0.4596~0.7605 |
| Jiangmen | -0.0057~0.0179 | 0.4537~0.5138 | 1.0839~1.3574 | 1.0839~1.3574 |
| Zhaoqing | -0.0057~0.0179 | 0.1756~0.2773 | 0.3285~0.4507 | 1.0839~1.3574 |

corresponding $R^2$ under OLS estimation was 0.5778, while the $R^2$ value under GWR model estimation was 0.7889, which shows that the latter results explain the variables with more explanatory power and better fit. And this result also confirms the aforementioned problem. The results of the regression coefficients for the four explanatory variables are shown in Table 10.

It can be seen that the factors influencing the coupling coordination are spatially heterogeneous, which further reflects the rationality of choosing the GWR model instead of the OLS model.

Agglomeration capacity has different directions (-0.0057–0.1007) and the least influence on the coupled coordination of urban quality and technology innovation. The regression coefficients show a spatially hierarchical band distribution with higher values in the central and eastern parts and lower values in the west. The areas with high coefficient values are concentrated in Guangzhou, Huizhou, Dongguan, and other cities, as the capital city of Guangdong Province and the core city of the Guangdong-Hong Kong-Macao Greater Bay Area, Guangzhou has a better foundation for economic development, a deep cultural richness, and a stronger attraction to labor and immigrants, which has radiated and led to the formation of high-value industrial clusters in the eastern part of the country, such as Huizhou and Dongguan. These cities need to introduce suitable population attraction policies and safeguards, explore new paths of agglomeration space governance, and further strengthen the role of agglomeration capacity in promoting coupling coordination.

Human capital has a more obvious positive feedback effect on the level of regional coupling coordination (0.1756–0.6397), and the regression coefficient shows a pattern of highs in the middle and lows in the fourth. It indicates that in cities with better development levels, the promotion effect of human capital on the coupling and coordination of urban quality and technology innovation is more obvious, the industrial structure of cities with high economic development levels is more advanced, which has a stronger need for labor quality and knowledge skill level, which is conducive to promoting the double-effect improvement of knowledge transfer and technology transformation and efficiently promoting the integration and co-progression of urban quality and technology innovation. Supported by policy dividends, Guangzhou, Zhongshan, and Zhuhai have experienced rapid economic and social development in recent years, which has created a higher demand for human capital.

Cultural development has a more obvious positive feedback effect on the level of regional coupling and coordination (0.3285–1.277), and the regression coefficient shows a pattern of

distribution that gradually decreases from the northeast to the southwest. Relatively speaking, cities with better economic development levels have higher-level requirements for spiritual and cultural life, and cultural development can better meet people's "soft needs," such as security, happiness, and fulfillment, and has a more obvious driving effect on the coupling coordination degree of urban quality and technology innovation. Culture is an important link, compared to the relatively weak economic and social development strength of the western part of the Guangdong-Hong Kong-Macao Greater Bay Area, the eastern part of the region more effectively played out in the construction of humanities Bay Area in the identity of the cohesion, the role of innovation. In particular, Guangzhou, as an example of a culturally strong city, can effectively stimulate the inherent driving force of cultural industry to promote the coupled and coordinated development of urban quality and technology innovation.

Infrastructure has a positive correlation (0.0815–1.3574) with the coupled and coordinated development of urban quality and technology innovation, and the spatial divergence effect is prominent, showing a spatial distribution pattern of high from the west and low from the east. It can be found that the area with the largest regression coefficient overlaps with the low value of the coupled and coordinated agglomeration area, indicating that for the disadvantaged cities, promoting the improvement of infrastructure can significantly promote the coupled and coordinated development of urban quality and technology innovation, strengthening the radiation effect of the core city on itself. Zhaoqing, Foshan, Jiangmen, although in Guangzhou, Macao and other core cities driven by the continuous development, but their own infrastructure strength is still insufficient, improve infrastructure development will provide more powerful support and protection of economic and social development. While Guangzhou's infrastructure development has been more complete, the driving effect is relatively weak.

## Conclusions

This paper explores the spatial and temporal evolution and driving mechanism of the coupling coordination of urban quality and technology innovation evaluation in the Guangdong-Hong Kong-Macao Greater Bay Area from 2010 to 2019 based on the urban quality and technology innovation evaluation index system, using the CRITIC-Entropy method, the coupling coordination development model, the geographic detector model, etc. The key conclusions are as follows:

(1)From 2011 to 2020, the spatial distribution characteristics of coupling coordination are "high in the middle and low around," with a logarithmic curve shape of "high in the east and low in the west" in the east-west direction, and an inverted U-shaped parabolic shape in the north-south direction.(2)The gravity center of coupling coordination shifts to the northeast direction from 2011 to 2020, and the unbalanced development trend of the north-south coupling coordination gradually eases. The standard deviation ellipse moved in the northeast direction from 2011 to 2020, showing a contraction trend away from the southwest direction. (3)Agglomeration capacity, human capital, cultural development, and infrastructure can greatly promote the improvement of the linking coordination of urban quality and technology innovation. The two-factor influence after the interaction is much higher.(4)The feedback effects of the coupling coordination states of different cities on each factor have obvious geographical differences, mainly showing the development trend of hierarchical band distribution.

The study on the spatial structure and driving mechanism of the linking coordination of urban quality and technology innovation can assist provide useful insights for the synergistic development of the Guangdong-Hong Kong-Macao Greater Bay Area as a whole. Based on the foregoing findings, this research puts forward the following optimization proposals:

1. Axis support and pole-axis radiation to facilitate regional integration building. Based on the "Guangdong-Shenzhen-Hong Kong-Macao Science and Technology Innovative Corridor," we will give full play to its position as an innovation growth pole and continue to implement the development notion of a "pole-led, axis-supported, radiated periphery." Build a regional coordination and multi-level linkage synergistic development mechanism.

2. Proactively create stable and coordination development ties with the coupling coordination of low-value agglomerations to break through the locking effect of low-value areas. the Pearl River. Based on the coupled and coordinated overall development of the Guangdong-Hong Kong-Macao Greater Bay Area, avoiding the problems of over-concentration and imbalance within the region.

3. Strengthen policy coordination and governance and improve policy-driven efficiency. Plays the role of regional coordination with the aid of macro regulation and control. Actively plays the active and dynamic role of the government, promotes the reason for a constructible allocation, high-quality coordination, and efficiency of various driving factors through the cohesion of various factors.

4. Insist on market demand as the guide and maximize the effectiveness of policies according to local conditions. In response to the spatial heterogeneity of connected cities, each city administration should identify its own weaknesses and execute measures appropriate to local conditions.

## Acknowledgments

Thank you to all the researchers, experts and editors who have helped us with our manuscript.

## Author Contributions

**Data curation:** Yixuan Wang.

**Formal analysis:** Yuxi Wu, Aichun Chen.

**Funding acquisition:** Fangfang Wang.

**Investigation:** Yixuan Wang.

**Methodology:** Zhichen Yang, Aichun Chen.

**Project administration:** Fangfang Wang.

**Supervision:** Yuxi Wu, Fangfang Wang.

**Writing – original draft:** Zhichen Yang, Yuxi Wu.

**Writing – review & editing:** Zhichen Yang, Yuxi Wu.

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
