## [Decision Letter · Decision Letter 0]

21 Mar 2023

PONE-D-23-05180Spatial-temporal differences and influencing factors of coupling coordination between urban quality and technological innovation in the Guangdong-Hong Kong-Macao Greater Bay AreaPLOS ONE

Dear Dr. Wu,

Thank you for submitting your manuscript to PLOS ONE. After careful consideration, we feel that it has merit but does not fully meet PLOS ONE’s publication criteria as it currently stands. Therefore, we invite you to submit a revised version of the manuscript that addresses the points raised during the review process.

We look forward to receiving your revised manuscript.

Kind regards,

Shi Yin

Academic Editor

PLOS ONE

Journal Requirements:

3. We note that Figure 1 and 4 in your submission contain [map/satellite] images which may be copyrighted. All PLOS content is published under the Creative Commons Attribution License (CC BY 4.0), which means that the manuscript, images, and Supporting Information files will be freely available online, and any third party is permitted to access, download, copy, distribute, and use these materials in any way, even commercially, with proper attribution. For these reasons, we cannot publish previously copyrighted maps or satellite images created using proprietary data, such as Google software (Google Maps, Street View, and Earth). For more information, see our copyright guidelines: http://journals.plos.org/plosone/s/licenses-and-copyright.

1. You may seek permission from the original copyright holder of Figure 1 and 4 to publish the content specifically under the CC BY 4.0 license.  

Natural Earth (public domain): " ext-link-type="uri" xlink:type="simple">http://www.naturalearthdata.com/"

Reviewers' comments:

Reviewer's Responses to Questions

**Comments to the Author**

1. Is the manuscript technically sound, and do the data support the conclusions?

Reviewer #1: Yes

Reviewer #2: Yes

Reviewer #3: Yes

2. Has the statistical analysis been performed appropriately and rigorously? 

Reviewer #1: Yes

Reviewer #2: No

Reviewer #3: Yes

3. Have the authors made all data underlying the findings in their manuscript fully available?

Reviewer #1: No

Reviewer #2: Yes

Reviewer #3: Yes

4. Is the manuscript presented in an intelligible fashion and written in standard English?

Reviewer #1: No

Reviewer #2: No

Reviewer #3: No

5. Review Comments to the Author

Reviewer #1: Dear Authors:

Taking the Guangdong-Hong Kong-Macao Greater Bay Area as an example, this paper analyzes the temporal and spatial differences and influencing factors of the coupling and coordination of urban quality and technological innovation. The research topic has research significance. The research method of the article is scientific, and the analysis is detailed. These are worthy of recognition, but there are also some problems, which are put forward for the author's reference:

1 The biggest problem in this article is the lack of theoretical analysis. The article lacks a theoretical analysis mechanism for the relationship between urban quality and technological innovation. That is, before quantitative analysis, it is necessary to construct how the relationship between urban quality and technological innovation affects. The above leads to the following quantitative analysis. It is recommended to add this part

2 The analysis in the quantitative results part lacks theoretical support, and it is suggested to supplement it.

3 The article did not carry out descriptive statistics on the variables used, including maximum value, minimum value, sample size, etc.

4 This article lacks a discussion section, which explains the marginal contribution and research limitations of the article.

5 The language section needs revision.

Kind regards

Reviewer #2: There are still many obvious problems in the paper, especially in the content style, which needs to be greatly revised.

First of all, the third part of the paper, the research results, does not give the research results, but only describes the calculation process, and the results is presented in the fourth part.

Secondly, the fourth part is not a real paper discussion, but now it is mainly the result of the research, and the discussion needs in-depth consideration.

Third, the conclusion of the fifth part is too much and needs to be condensed.

Fourth, the details of the article still need to be carefully revised, such as punctuation, such as the writing of digital units.

Reviewer #3: I found the paper entitled ‘’Spatial-temporal differences and influencing factors of coupling coordination between urban quality and technological innovation in the Guangdong-Hong Kong-Macao Greater Bay Area’’ valuable, since it measures the coupling relationship between urban quality and science and technology innovation using the coupling coordination model, explores the state characteristics and spatial structure of the linkage between the two using trend surface analysis and standard deviation ellipse analysis. While this article has enough value to be published, there are some issues that to be addressed: as follows.

(1) Page 3, Paragraph 1, Line 3-5: ‘’In 2021, China's urbanization rate will have reached 64.72%’’. Please check the tense and grammar of this sentence.

(2) Page 3, Paragraph 2, Line 5: ‘’are of great significance to deepening the cooperation’’. Please check the grammar of this sentence.

Please have your paper being edited by a native English speaker who is either your friend or from an Editing company.

(3) Page 7, Paragraph 2, Line 1: The definition of entropy needs to be cited.

(4) Page 8: Correct the denominator of the formula (2).

(5) Page 10, Paragraph 3: Why the values of a and b were set as 0.5?

(6) Page 14, Paragraph 2: How the trend surface is obtained? If it was obtained by the least squares approach, how are the parameters of formula (14) determined? How to determine the parameters are the most suitable?

(7) Page 17, Paragraph 1, Line 6:’’OLS’’ should be defined for the first time in the paper.

(8) Page 17, Paragraph 1: It’s better to add a figure to compare GWR model and OLS model.

(9) Page 20, Paragraph 2: Why the q value of human capital is the lowest?

(10) Page 21, Paragraph 1, Line 15: What are the three industries?

(11) Page 31-32: There are two Figure 4. Furthermore, the quality of second Figure 4 should be increased. Legends are not readable.

6. PLOS authors have the option to publish the peer review history of their article (what does this mean?). If published, this will include your full peer review and any attached files.

Reviewer #1: **Yes: **Anlu Zhang

Reviewer #2: No

Reviewer #3: No

---

## [Author Response · Author response to Decision Letter 0]

5 May 2023

Many thanks to the reviewers and editors who took the time to review the manuscript and provided very encouraging comments on its merits. We have made careful revisions to the manuscript. We are particularly grateful to the editors for their reference suggestions on the format of information and copyright of maps in our manuscript, and we have made all the changes in the manuscript. We have uploaded the responses in word format to the system, and the file names are "Response to Reviewer 1" "Response to Reviewer 2" and "Response to Reviewer 3.".

Once again, we would like to thank all the reviewers and editors for their careful reviews and comments.

Best wishes

All Authors

---

## [Decision Letter · Decision Letter 1]

4 Jun 2023

PONE-D-23-05180R1Spatial-temporal differences and influencing factors of coupling coordination between urban quality and technological innovation in the Guangdong-Hong Kong-Macao Greater Bay AreaPLOS ONE

Dear Dr. Wu,

Thank you for submitting your manuscript to PLOS ONE. After careful consideration, we feel that it has merit but does not fully meet PLOS ONE’s publication criteria as it currently stands. Therefore, we invite you to submit a revised version of the manuscript that addresses the points raised during the review process.

If applicable, we recommend that you deposit your laboratory protocols in protocols.io to enhance the reproducibility of your results. Protocols.io assigns your protocol its own identifier (DOI) so that it can be cited independently in the future. For instructions see: https://journals.plos.org/plosone/s/submission-guidelines#loc-laboratory-protocols. Additionally, PLOS ONE offers an option for publishing peer-reviewed Lab Protocol articles, which describe protocols hosted on protocols.io. Read more information on sharing protocols at https://plos.org/protocols?utm_medium=editorial-emailutm_source=authorlettersutm_campaign=protocols.

We look forward to receiving your revised manuscript.

Kind regards,

Shi Yin

Academic Editor

PLOS ONE

Reviewers' comments:

Reviewer's Responses to Questions

**Comments to the Author**

1. If the authors have adequately addressed your comments raised in a previous round of review and you feel that this manuscript is now acceptable for publication, you may indicate that here to bypass the “Comments to the Author” section, enter your conflict of interest statement in the “Confidential to Editor” section, and submit your "Accept" recommendation.

Reviewer #3: All comments have been addressed

Reviewer #4: (No Response)

Reviewer #5: (No Response)

Reviewer #6: All comments have been addressed

2. Is the manuscript technically sound, and do the data support the conclusions?

Reviewer #3: Yes

Reviewer #4: Yes

Reviewer #5: No

Reviewer #6: Partly

3. Has the statistical analysis been performed appropriately and rigorously? 

Reviewer #3: Yes

Reviewer #4: Yes

Reviewer #5: N/A

Reviewer #6: Yes

4. Have the authors made all data underlying the findings in their manuscript fully available?

Reviewer #3: Yes

Reviewer #4: Yes

Reviewer #5: No

Reviewer #6: Yes

5. Is the manuscript presented in an intelligible fashion and written in standard English?

Reviewer #3: Yes

Reviewer #4: Yes

Reviewer #5: No

Reviewer #6: Yes

6. Review Comments to the Author

Reviewer #3: No problem, you can revise the sentence logic again，With regard to the issues raised by the reviewers, the authors have made detailed changes

Reviewer #4: The authors investigate the coupling coordination relationship between urban quality and technological innovation in GBA. This work investigates them from entropy TOPSIS, coupling coordination models, the gravity center and standard deviation ellipse method, the geographic probe, the GWR, and other methods method. The method is propriated. However, there are a few details to be considered.

1. The language section still needs revision. Such as line 450, 453, and etc. I suggest the authors check the manuscript carefully.

2. There are several papers study the sustainable development in GBA, such as https://doi.org/10.3846/tede.2022.16618, https://doi.org/10.1111/grow.12636. It should include in the manuscript.

Reviewer #5: Dear Author/Editor:

Based on entropy TOPSIS, coupling coordination models, and other methods, this paper explored the spatial variation and influencing factors of the coupling coordination relationship between urban quality and technology innovation in the Guangdong-Hong Kong-Macao Greater Bay Area from 2011 to 2020. The article is relatively innovative in its research perspective, extensive in its content, and clear in its graphical presentation, but several important issues affect its publication:

1）The article has a large number of grammatical problems and unclear expressions, for example in the methodology section and the overview of the study area.

2) The article's theoretical value and practical significance are not significant, and the coupling relationship between urban quality and science and technology innovation is not very clearly expressed.

3)Article title and main content need to be consistent. For instance， “technological innovation”、“Science and technology innovation”.

4) The data sources are not clearly expressed in part, and some data need to be supplemented with specific websites and references. Also, how are the data for Macau and Hong Kong handled? There is the problem of exchange rate conversion.

5) The indicator system of urban quality and technological innovation is not explained clearly. Some important references are missing. And the indicator system needs to be by the characteristics and actual situation of the Guangdong-Hong Kong-Macao Greater Bay Area.

6)The results analysis section is relatively weak, and no in-depth excavation has been carried out. In particular, there is no summary and practical analysis of the coupling characteristics and influencing factors of urban quality and technological innovation.

7) There are too many words in the discussion section, there are some contents that can be put inside the result analysis.

Reviewer #6: The authors have made revisions to the article as required, and the overall level of publication has been reached

7. PLOS authors have the option to publish the peer review history of their article (what does this mean?). If published, this will include your full peer review and any attached files.

Reviewer #3: No

Reviewer #4: No

Reviewer #5: No

Reviewer #6: No

---

## [Author Response · Author response to Decision Letter 1]

19 Jun 2023

We would like to thank all the reviewers for their careful review and responses, and we have uploaded the specific responses returned to the reviewers as an attachment. Thank you

---

## [Decision Letter · Decision Letter 2]

4 Jul 2023

PONE-D-23-05180R2Spatial-temporal differences and influencing factors of coupling coordination between urban quality and technology innovation in the Guangdong-Hong Kong-Macao Greater Bay AreaPLOS ONE

Dear Dr. Wang,

Thank you for submitting your manuscript to PLOS ONE. After careful consideration, we feel that it has merit but does not fully meet PLOS ONE’s publication criteria as it currently stands. Therefore, we invite you to submit a revised version of the manuscript that addresses the points raised during the review process.

If applicable, we recommend that you deposit your laboratory protocols in protocols.io to enhance the reproducibility of your results. Protocols.io assigns your protocol its own identifier (DOI) so that it can be cited independently in the future. For instructions see: https://journals.plos.org/plosone/s/submission-guidelines#loc-laboratory-protocols. Additionally, PLOS ONE offers an option for publishing peer-reviewed Lab Protocol articles, which describe protocols hosted on protocols.io. Read more information on sharing protocols at https://plos.org/protocols?utm_medium=editorial-emailutm_source=authorlettersutm_campaign=protocols.

We look forward to receiving your revised manuscript.

Kind regards,

Shi Yin

Academic Editor

PLOS ONE

Reviewers' comments:

Reviewer's Responses to Questions

**Comments to the Author**

1. If the authors have adequately addressed your comments raised in a previous round of review and you feel that this manuscript is now acceptable for publication, you may indicate that here to bypass the “Comments to the Author” section, enter your conflict of interest statement in the “Confidential to Editor” section, and submit your "Accept" recommendation.

Reviewer #7: (No Response)

Reviewer #8: All comments have been addressed

2. Is the manuscript technically sound, and do the data support the conclusions?

Reviewer #7: Yes

Reviewer #8: Yes

3. Has the statistical analysis been performed appropriately and rigorously? 

Reviewer #7: Yes

Reviewer #8: Yes

4. Have the authors made all data underlying the findings in their manuscript fully available?

Reviewer #7: Yes

Reviewer #8: Yes

5. Is the manuscript presented in an intelligible fashion and written in standard English?

Reviewer #7: Yes

Reviewer #8: Yes

6. Review Comments to the Author

Reviewer #7: The manuscript conducts a series of analysis on the subject of the coupling coordination between urban quality and technology innovation in the Guangdong-Hong Kong-Macao Greater Bay Area. The topic is interesting and worthy of research. The manuscript does extensive research in terms of content, but I still have several concerns which are listed in the following.

1. The time duration on the study is from 2011 to 2020. Are there any specific reasons for choosing this duration?

2. The manuscript does a bunch of analysis, but the structure or relationship between the analysis is not clearly described. As a result, it seems that the analysis is combined by a series of pieces with weak connections. It is better to have a figure to present the logic structure on all the analysis.

3. Some of the analysis results are lack of suitable discussions on their implications and reasons. For example, in the analysis of spatial heterogeneity based on GWR, most results shown in the manuscript are the statistical analysis. The authors are expected to discuss why the coefficients estimated from the models are different for different regions.

Reviewer #8: (1)It is recommended to cite some references from 2023.

(2)It is recommended to beautify Figure 1.

(3)The description of the study area should be as brief as possible

7. PLOS authors have the option to publish the peer review history of their article (what does this mean?). If published, this will include your full peer review and any attached files.

Reviewer #7: No

Reviewer #8: No

---

## [Author Response · Author response to Decision Letter 2]

17 Jul 2023

Dear Reviewer,

Thank you very much for your time involved in reviewing the manuscript and your very encouraging comments on the merits. We have carefully revised the manuscript.

Comments: 

“The manuscript conducts a series of analysis on the subject of the coupling coordination between urban quality and technology innovation in the Guangdong-Hong Kong-Macao Greater Bay Area. The topic is interesting and worthy of research. The manuscript does extensive research in terms of content, but I still have several concerns which are listed in the following.”

Response :

Thank you very much for recognizing our manuscript and acknowledging the value of the research content. We also appreciate your clear and detailed feedback and hope that the explanation has fully addressed all of your concerns. In the remainder of this letter, we discuss each of your comments individually along with our corresponding responses.

To facilitate this discussion, we first retype your comments in italic font and then present our responses to the comments. 

(1) The time duration on the study is from 2011 to 2020. Are there any specific reasons for choosing this duration?

Response(1) :

Thank you for the detailed review. The choice of 2011-2020 is mainly based on the formal establishment and development of the Guangdong-Hong Kong-Macao Greater Bay Area. In 2014, the Shenzhen Municipal Government Work Report proposed to "build a Bay Area economy", and in July 2017, Guangdong, Hong Kong, and Macau formally signed the "Proposal on the Framework for Deepening Co-operation among Guangdong, Hong Kong, and Macao and Promoting the Construction of the Greater Bay Area", and in early 2019, the Central Government formally issued the "Outline of the Plan for the Development of the Greater Bay Area of Guangdong, Hong Kong and Macao". The ten-year period from 2011-2020 is a landmark decade in the development of the Greater Bay Area of Guangdong, Hong Kong, and Macao from a local policy to a national strategy, and encompasses the three important points in the development of the Greater Bay Area of Guangdong, Hong Kong and Macao, which is more representative of the continuous progress and rapid changes in the quality of the cities in the Bay Area as well as in science and technology innovations. It is more representative of the continuous progress and rapid changes in the quality of cities and technological innovation in the Bay Area.

(2) The manuscript does a bunch of analysis, but the structure or relationship between the analysis is not clearly described. As a result, it seems that the analysis is combined by a series of pieces with weak connections. It is better to have a figure to present the logic structure on all the analysis.

Response(2) :

Thank you very much for your suggestion, it also makes us realize that it was not clearly expressed in our manuscrip. We have added a technology roadmap to the manuscript to hopefully provide a better understanding of what we are researching and thinking about.

(3) Some of the analysis results are lack of suitable discussions on their implications and reasons. For example, in the analysis of spatial heterogeneity based on GWR, most results shown in the manuscript are the statistical analysis. The authors are expected to discuss why the coefficients estimated from the models are different for different regions.

Response(3) : 

Thanks to your suggestion, we have added part of the discussion and analysis to the manuscript and have highlighted all of them in red.

(4)It is recommended to cite some references from 2023.

Response 4:

Thank you for your suggestion about our references, we agree with you very much and have updated the references in the manuscript。

(5)It is recommended to beautify Figure 1.

Response 5:

Thanks to your suggestions, We have updated Figure 1.

(6)The description of the study area should be as brief as possible.

Response 6:

Thank you for your suggestion, and after discussion, we have changed the study area overview section of the manuscript, which now reads as follows:

The Guangdong-Hong Kong-Macao Greater Bay Area is located in the coastal region of South China (111°21′~114°53′E, 21°28′~24°29′N), and consists of the two special administrative regions of Hong Kong and Macau and the nine Pearl River Delta(PRD) cities of Guangzhou, Shenzhen, Zhuhai, Foshan, Zhongshan, Jiangmen, Huizhou, Dongguan, and Zhaoqing in Guangdong Province. The Guangdong-Hong Kong-Macao Greater Bay Area has a leading level of economic development, a complete industrial system, obvious cluster advantages, and the special advantages of "one country, two systems, and three customs zones," and has always been one of the regions with the highest degree of openness and economic vitality in China. In February 2019, the Central Committee of the Communist Party of China and the State Council issued "the outline of the development plan of the Guangdong-Hong Kong-Macao Greater Bay Area," making it clear that the Guangdong-Hong Kong-Macao Greater Bay Area will not only become a vibrant world-class city cluster, an international center of technology innovation, an important support for the construction of the Mainland, Hong Kong, and Macao in-depth cooperation demonstration area, but also a quality living circle that is pleasant to live, work, and travel, and become a model of high-quality development.

We would like to take this opportunity to thank you again for all your time involved and this great opportunity for us to improve the manuscript. We hope you will find this revised version satisfactory.

Sincerely,

Fangfang Wang

---

## [Decision Letter · Decision Letter 3]

31 Jul 2023

Spatial-temporal differences and influencing factors of coupling coordination between urban quality and technology innovation in the Guangdong-Hong Kong-Macao Greater Bay Area

PONE-D-23-05180R3

Dear Dr. Fangfang Wang,

We’re pleased to inform you that your manuscript has been judged scientifically suitable for publication and will be formally accepted for publication once it meets all outstanding technical requirements.

Kind regards,

C. A. Zúniga-González, Ph.D

Academic Editor

PLOS ONE

Additional Editor Comments (optional):

Dear author I am checked that you have addresses all reviewers' observations. My sincere congratulations!!!!!!!

Reviewers' comments:

Reviewer's Responses to Questions

**Comments to the Author**

1. If the authors have adequately addressed your comments raised in a previous round of review and you feel that this manuscript is now acceptable for publication, you may indicate that here to bypass the “Comments to the Author” section, enter your conflict of interest statement in the “Confidential to Editor” section, and submit your "Accept" recommendation.

Reviewer #1: All comments have been addressed

Reviewer #4: All comments have been addressed

Reviewer #5: All comments have been addressed

2. Is the manuscript technically sound, and do the data support the conclusions?

Reviewer #1: Yes

Reviewer #4: Yes

Reviewer #5: Yes

3. Has the statistical analysis been performed appropriately and rigorously? 

Reviewer #1: Yes

Reviewer #4: Yes

Reviewer #5: Yes

4. Have the authors made all data underlying the findings in their manuscript fully available?

Reviewer #1: Yes

Reviewer #4: Yes

Reviewer #5: Yes

5. Is the manuscript presented in an intelligible fashion and written in standard English?

Reviewer #1: Yes

Reviewer #4: Yes

Reviewer #5: Yes

6. Review Comments to the Author

Reviewer #1: Dear authors :

Thank you for your revise and the revised article is good and it is recommended to publish it

Kind regards

Reviewer #4: After careful consideration, we have no further comments and it is suitable for publication as it currently stands. Therefore, my decision is "ACCEPT."

Reviewer #5: The article has made some modifications, added the roadmap and framework of the article, and has a certain depth to the conclusion.

Agree to accept.

7. PLOS authors have the option to publish the peer review history of their article (what does this mean?). If published, this will include your full peer review and any attached files.

Reviewer #1: No

Reviewer #4: No

Reviewer #5: No

---

## [Editor Report · Acceptance letter]

13 Sep 2023

PONE-D-23-05180R3 

Spatial-temporal differences and influencing factors of coupling coordination between urban quality and technology innovation in the Guangdong-Hong Kong-Macao Greater Bay Area 

Dear Dr. Wang:

I'm pleased to inform you that your manuscript has been deemed suitable for publication in PLOS ONE. Congratulations! Your manuscript is now with our production department. 

Kind regards, 

on behalf of

Dr. Prof. C. A. Zúniga-González 

%CORR_ED_EDITOR_ROLE%

PLOS ONE